# Quality and costs of commissioned vs. researcher-led NIHR research: retrospective cohort study of randomised controlled trials

**Ashley Hammond**◉*, **Alastair D. Hay**

Centre for Academic Primary Care, NIHR School for Primary Care Research, Population Health Sciences, University of Bristol, Bristol, England

\* ashley.hammond@bristol.ac.uk

## Abstract

The NIHR provides two main funding pathways: 'commissioned' (CP) and 'researcher-led' (RLP) projects. CPs are used to ensure NHS priorities are addressed. However, there is uncertainty regarding which provides the highest quality, best value for money, research. We compared the quality and cost of commissioned and research-led projects. We searched the NIHR Funded Portfolio database for randomised controlled trials (RCTs) funded by programmes offering both CP and RLP funding. The main outcome was journal impact factor for the main results paper. Other outcomes included: Altmetric score, relative citation ratio, total and per participant funding. T-tests were used to compare mean values. 82 eligible RCTs were identified, 30 commissioned and 52 researcher-led. One RLP did not progress beyond pilot; two (one CP and one RLP) did not publish a monograph; and 78 published main study results in a peer-reviewed journal. Among these, RLP projects had higher means scores than CP projects for: journal impact factor (111.2 and 40.8, respectively, p < 0.001); Altmetric score (223.8 and 116.8, p = 0.05); and relative citation ratio (8.0 and 4.4, p = 0.06). Among the full 82 RCTs, RLP mean total study and per participant funding were lower than CP: £1,429,913 vs. £1,353,034, p = 0.59; and £11,271 vs. £4425, p = 0.08. Researcher-led projects are more likely to result in higher impact research compared with commissioned studies. Further research is needed to understand the reasons, which could include: commissioned research addressing harder-to-do questions; commissioning brief quality; and/or research team motivation.

## Introduction

The National Institute for Health and Care Research (NIHR) was founded in 2006 and now has an annual budget of £1.3 billion, and is the UK's main funder of health research, alongside other UK research funders including the Medical Research Council with an annual budget of over £800 million, and UK Research and Innovation with an annual budget of £6.2 billion. There are currently 17 NIHR funding programmes available, supporting patient, social care, public health, evidence synthesis, innovation and global health research.

All programmes invite the research community to submit applications addressing questions of their choice, so-called 'researcher-led' projects (RLPs). Sometimes these will be in

**Data availability statement:** The data underlying the results presented in the study are available from the publicly available NIHR funded Portfolio Database. Our dataset uploaded from this database and used in the analysis is held in on Figshare (https://figshare.com/articles/dataset/NIHR_Open_Data_-_Funded_Portfolio/28303580?file=52336322). Individual project details are all available from the NIHR Journals Library (https://www.journalslibrary.nihr.ac.uk/#/).

**Funding:** The author(s) received no specific funding for this work.

**Competing interests:** The authors have declared that no competing interests exist.

response to 'themed calls' designed to stimulate research into a broad area, such as anti-microbial resistance. RLPs are assessed by funding committees on their individual merit, usually without reference to other projects, since it would be usual for committees to see RLPs addressing similar questions.

Some programmes also invite 'commissioned' projects (CPs) by publishing 'commissioning briefs' designed to stimulate research to address specific research questions. Commissioning briefs often provide detailed descriptions of what is wanted in terms of population, intervention, comparator and outcomes (PICO). Responding researchers are advised to adhere closely to the brief because: (i) the research question has been vetted and deemed a priority; and (ii) the programme needs to compare competing bids.

While it is important priority questions are addressed, commissioning briefs also need to describe research questions that experts consider correct and feasible. Unfortunately, researchers may be conflicted, putting their desire to secure funding ahead of raising concerns about the brief. If this is occurring frequently, it could lead to poorer quality, lower value-for-money, research.

We therefore hypothesised that NIHR-funded researcher-led projects provide higher quality, better value-for-money research than commissioned projects.

## Methods

The NIHR Funded Portfolio Database was searched for eligible projects, which includes and identifies both commissioned and researcher-led research projects [1]. Projects meeting all the following inclusion criteria were extracted from the database: (i) NIHR-funded under any programme offering both commissioned and researcher-led opportunities, (ii) funding end date between 1$^{st}$ January 2017 and 31$^{st}$ December 2018 (to allow time for studies to complete their final analyses and publish their main study results), and (iii) two or multiple-arm, individually or cluster randomised, superiority or non-inferiority RCTs (so similar that we were only comparing experimental studies). Projects were excluded if they used non-randomised designs, were feasibility studies or systematic reviews of RCTs.

The eligible project IDs were then individually searched for in the NIHR Journals Library to extract the following data per project: [2] whether the project progressed beyond the initial pilot phase (yes/no), monograph published (yes/no) and number of monograph pages, publication of main study results (yes/no). The NIHR Journals Library include six open-access journals providing information about research funded by the NIHR, and typically include a monograph of each project. Monographs have a higher word limit (approximately 50,000) and so therefore allow the addition of significantly more detail than a paper published in a peer-reviewed journal [3]. To determine impact, our main outcome measure was the impact factor of journal where main results were published. Journal impact factor data was collected for all eligible projects with published results, and was collected in August 2024, as unfortunately it was not possible to collect journal impact factor data at the time of publication for each included published paper, due to lack of data availability. We also collected the Altmetric score for every published project via the Altmetric website [4]. The Altmetric score is a measure of the online attention a research output receives, including mainstream and social media, and also government and public policy documents. It provides an effective online tool to understand who is engaging with research online, and is a useful measure of the overall influence of a research project. Finally, Relative Citation Ratio (RCR) data was also collected for all eligible published projects, which is typically available via the paper metrics on the journal site or via the linked research database Dimensions [5]. The RCR is a citation-based measure of the influence of a research output, and is calculated by the number of citations

a published paper receives, normalised to the citations received by other publications in the same area of research and year [6]. The area is dynamically defined by the collection of publications co-cited with each published paper (the co-citation network) [6]. The RCR is calculated for all published papers available on PubMed which are at least two years old, and are centred around 1.0, meaning that a paper with an RCR of 1.0 has received the same number of citations as would be expected based on other published papers in that research area that year, whereas a paper with an RCR of 2.0 has twice as many citations as would be expected.

Baseline data for all variables were reported, including the name of the funding programme, RCT design, and progression beyond the initial pilot phase; grouped by type of funding (commissioned and researcher-led). For our main outcome, impact factor of the journal where main results are published, student's t-tests were used (provided the data satisfied all student's t-test assumptions) to determine the relationship between impact factor and type of funding. Student's t-tests were also run for Altmetric score, RCR and total and per participant costs (provided the data satisfied all student's t-test assumptions).

## Patient and public involvement

No patients or members of the public were involved in the study as we did not collect data directly from individuals.

## Results

Our search identified RCTs within four NIHR funding programmes which offer both commissioned and researcher-led opportunities: Efficacy and Mechanism Evaluation (EME), Health Technology Assessment (HTA), Health and Social Care Delivery Research (HSCDR), and Public Health Research (PHR). In total 298 projects were identified across the four funding programmes with an end date between 2017 and 2018 calendar years, of which 82 were eligible RCTs (Table 1). All 82 projects were searched for in the NIHR journals library, which linked to the published monograph (available for 80/82 projects), and to the publication of the main study results. However, for 63/82 projects, the final published manuscript had to be searched for manually using the study title in PubMed search engine.

We found strong evidence of RLPs being published in higher impact factor journals than CPs (Table 2). The mean journal impact factor for CPs was 40.8 (95% confidence interval 20.1 to 61.6), versus 111.2 (95% CI 89.9 to 132.5) for RLPs, p < 0.001. Altmetric score was also higher for RLPs compared with CPs, with a mean of 223.8 (95% CI 151.6 to 296.1) and 116.8 (95% CI 44.6 to 190.0), respectively, with marginal statistical significance at p = 0.05. Relative citation ratio was also higher for RLPs compared with CPs, with a mean of 8.0 (95% CI 5.2 to 108) and 4.4 (95% CI 2.9 to 6.0), respectively, p = 0.06.

The mean award amount in CP was not found to be significantly different to RLPs, shown below in Table 3, at £1,429,913 and £1,353,034, respectively, p = 0.59. There was little evidence of a difference in mean cost per participant between CPs and RLPs (Table 3), at £11,271 and £4425, respectively, p = 0.08.

## Discussion

### Summary of main findings

This study provides strong evidence that researcher-led projects are published in higher impact factor journals compared to commissioned projects. To our knowledge, this is the first time such an analysis has been published and if verified, raises important questions for how commissioned projects are developed and delivered.

Table 1. Baseline data for commissioned vs. researcher-led NIHR funded RCTs.

| | Type of funding | |
|---|---|---|
| | Commissioned | Researcher-led |
| Total number of eligible RCTs with end date between 2017 and 2018 | 30 | 52 |
| Programme: | | |
| Efficacy and Mechanism Evaluation | 1 | 10 |
| Health Technology Assessment | 26 | 40 |
| Health and Social Care Delivery Research | 1 | 1 |
| Public Health Research | 2 | 1 |
| RCT design: | | |
| Superiority | 28 | 47 |
| Non-inferiority | 2 | 5 |
| Progression beyond initial pilot phase? | | |
| Yes | 30 | 51 |
| No | 0 | 1 |
| NIHR monograph published? | | |
| Ye s | 29 | 51 |
| No | 1 | 1 |
| Main study results published in non-NIHR journal? | | |
| Yes | 29 | 49 |
| No | 1 | 3 |

Table 2. Relationship between impact factor of publishing journal and type of funding for 78 eligible studies with published results.

| | Commissioned | Researcher-led | p-value (Student's t-test) |
|---|---|---|---|
| Impact factor of journal where main results are published | | | |
| Mean | 40.8 | 111.2 | <0.001 |
| 95% confidence interval | 20.1 to 61.6 | 89.9 to 132.5 | |
| Altmetric score | | | |
| Mean | 116.8 | 223.8 | 0.05 |
| 95% confidence interval | 44.6 to 190.0 | 151.6 to 296.1 | |
| Relative Citation Ratio | | | |
| Mean | 4.4 | 8.0 | 0.06 |
| 95% confidence interval | 2.9 to 6.0 | 5.2 to 10.8 | |

## Strengths and weaknesses

Strengths of this study include its originality, sample size sufficiency to address the question, and objectivity of outcomes. We are not aware of any existing research on this topic to date, which highlights the importance of our findings and the need for further research.

The main weakness of this study is that there is no single accepted measure of research quality. Many metrics are available, measuring different research output domains. To mitigate the inherent subjectivity in quality assessment of research outputs, we used two compensatory strategies: we decided *a priori* that journal impact factor would be our main quality measure; and we decided to report several measures. Impact factor of the journal where the main results

**Table 3. Total amount awarded and mean costs per participant recruited for 78 studies with published results.**

| | Commissioned | Researcher-led | p-value (Student's t-test) |
|---|---|---|---|
| **Total amount awarded** | | | 0.59 |
| **Mean** | £1,429,913 | £1,353,034 | |
| **95% confidence interval** | £1,183,130–£1,676,696 | £1,184,867–£1,521,202 | |
| **Mean cost per participant recruited** | | | 0.08 |
| **Mean** | £11,271 | £4425 | |
| **95% confidence interval** | £1417–£21,124 | £2246 - £6604 | |

are published acts as a proxy for the quality of the journal, and is directly related to citations; the more citations a journal has, the higher its impact factor [7]. However this measure does not directly reflect the quality of the individual research output. Altmetric scores are a measure of the broader influence of a research article, focused on the online attention an article receives. It provides a measure of how often others are accessing and using it worldwide, what kind of online discussions the article is generating, and how it may be affecting policy. Altmetric scores are designed to complement citation-based measures such as impact factor by providing non-traditional article-level research metrics that reflect the way information is typically processed today [8]. Relative citation ratio (RCR) provides a measure of within field-of-research quality, allowing comparison of papers addressing differing health problems. For example, a research paper with an RCR of 4.0 indicates that, on average, the article was four times more likely to be cited than comparable papers within the same field published the same year [6]. We believe that using these compensatory strategies was able to enhance our overall assessment of research quality, and our findings were consistent across all selected quality measures.

Future quality assessments could include other more direct impact measures such as implementation of public policies and increasing the population health demand coverage being addressed.

It must also be considered that commissioned projects are likely to be addressing very specific research questions within under-researched areas, for example neglected diseases, which are less commonly researched but still reflect an important public health issue. This highlights that there are likely important differences with respect to the topic and field of research being funded by commissioned and researcher-led projects.

## Implications

Research is needed to understand why these differences are occurring. We speculate that there are three contributory factors. First, commissioned projects may seek to address harder-to-do research questions that the research community considers too difficult or at high risk of 'failure'. We argue there should be mechanisms that acknowledge and reward researchers whose projects both 'succeed' and 'fail' and that do not penalise researchers willing to address difficult-to-do research. To our knowledge, no research has been conducted to date to try to address these implications. Second, funders need to ensure commissioning briefs are adequately peer-reviewed for importance, clinical relevance and feasibility. Our experience is that projects are more likely to succeed when patients and those supporting recruitment regard the question as relevant and important. Finally, maintaining the motivation of a research team to take a project through to completion, often over many years and before results are seen, is easier when the research question is widely seen as important.

## Conclusions

Researcher-led projects are more likely to result in higher impact research compared with commissioned studies. Research is needed to understand the reasons, which could include: commissioned research addressing harder-to-do questions; commissioning brief quality; and/ or research team motivation.

## Acknowledgements

We acknowledge the support provided by Professor John Norrie, Chair of MRC/NIHR Efficacy and Mechanism Evaluation Board, and Professor Danny McAuley, Scientific Director for NIHR Programmes, who both contributed to early discussions of research methodologies, and reviewed early versions of the manuscript.

## Author contributions

**Conceptualization:** Alastair D Hay.

**Data curation:** Ashley Hammond.

**Formal analysis:** Ashley Hammond.

**Investigation:** Ashley Hammond, Alastair D Hay.

**Methodology:** Ashley Hammond, Alastair D Hay.

**Resources:** Ashley Hammond.

**Writing – original draft:** Ashley Hammond.

**Writing – review & editing:** Ashley Hammond, Alastair D Hay.

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
