## [Decision Letter · Decision Letter 0]

3 Jan 2025

PONE-D-24-39387Quality and costs of commissioned vs. researcher-led NIHR research: retrospective cohort study of randomised controlled trialsPLOS ONE

Dear Dr. Hammond,

Thank you for submitting your manuscript to PLOS ONE. After careful consideration, we feel that it has merit but does not fully meet PLOS ONE’s publication criteria as it currently stands. Therefore, we invite you to submit a revised version of the manuscript that addresses the points raised during the review process.

In your review, please pay careful attention to each comment made by Reviewers 1 and 2, providing detailed information on the screening of the database, and making the strategies for inter-rater reliability clearer. In the section Strengths and Weaknesses,you statet that "[t]he main weaknesses [please correct typo] of this study is that there is no single accepted measure of research quality". You explain that you used two compensatory strategies and then provided Altmetric scores and relative citation ratio to avoid limiting your analysis to the JIF. You then report that the "findings were consistent across all selected quality measures". Using these compensatory strategies to manage the inherent subjectivity in quality assessment seems to be framed as negative. However, you used these strategies to enhance the overall assessment of quality. Please, clarify.

We look forward to receiving your revised manuscript.

Kind regards,

Sonia Vasconcelos, PhD

Academic Editor

PLOS ONE

Journal Requirements:

3. We note you have included a table to which you do not refer in the text of your manuscript. Please ensure that you refer to Table 3 in your text; if accepted, production will need this reference to link the reader to the Table.

Reviewers' comments:

Reviewer's Responses to Questions

**Comments to the Author**

1. Is the manuscript technically sound, and do the data support the conclusions?

Reviewer #1: Yes

Reviewer #2: Partly

2. Has the statistical analysis been performed appropriately and rigorously? 

Reviewer #1: Yes

Reviewer #2: Yes

3. Have the authors made all data underlying the findings in their manuscript fully available?

Reviewer #1: No

Reviewer #2: Yes

4. Is the manuscript presented in an intelligible fashion and written in standard English?

Reviewer #1: Yes

Reviewer #2: Yes

5. Review Comments to the Author

Reviewer #1: The link with supplementary data did not open.

Maybe CP projects were about certain topic as neglected diseases, or diseases that are not prone to commercial interests but that are important as a public health issue. This should be included at Discussion topic. For example, in 2021, covid studies were by far more cited than others in health area.

In the discussion, it was mentioned that there are no established criteria to measure the quality of research. However, another topic that could be addressed in the debate is the implementation of public policies and the increase in population coverage to address a specific health demand; these could also be used as quality indicators. I understand that conducting these analyses was not the objective of the study, but a paragraph reflecting on this topic should be included in the Discussion.

Considering the importance of the funds invested in promoting health research in the United Kingdom, this work is extremely relevant to the discussion on the efficient use of financial resources, given that demand will always exceed the population's health needs. In a context of scarce resources, addressing this topic is highly significant.

Reviewer #2: This is an interesting and important analysis of funding health research. There are several issues that need to be clarified. My comments are in order of their appearance in the manuscripts, not in order of their importance.

1. Abstract - the Results section stated that there were some differences in the parameters measured, but the P value i not significant (there is one that is 0.05, but that should not be considered significant). The authors do not make such statements in the manuscript body.

2. Introduction: More information should be provided for RPLs and CPs in terms of the length of the proposal, maximum amount of the grant, number of researchers?

4. Methods: The screening of the database and extractions of data - was it performed independently by the two authors? Was there any attempt to assess the agreement between them?

5. Methods: The monographs should be described in more detail. Are they articles like the published articles?

6. Methods: The calculation of RCR is not clearly explained. How was the area of research determined for individual projects? How many were such areas?

7. Methods: Did the authors check the normality of data distribution, considering that the sample sizes were not big?

8. Results, table 1: Did all projects had a monograph and a paper published? What is the difference? Also, what is a non-NIHR journal? NIHR seems to publish monographs and the difference is not well explained.

9. Results, table 2: It is not clear when the papers were published to make sense of the impact factor data.

11. Results, table 2: The statistical test should be indicated in the table (Student t-test).

12. Results, table 2: The authors state there were differences in parameters, but only the difference between journal impact factors was significant.

13. Results, table 3: The authors state that "The mean award amount in CP was only marginally higher" but the p value was 0.59.

14. Discussion: The main conclusions of the study should be based only on results with clear statistical significance.

6. PLOS authors have the option to publish the peer review history of their article (what does this mean? ). If published, this will include your full peer review and any attached files.

**Do you want your identity to be public for this peer review?** For information about this choice, including consent withdrawal, please see our Privacy Policy .

Reviewer #1: **Yes: ** Maria Gabriela Haye Biazevic

Reviewer #2: No

---

## [Author Response · Author response to Decision Letter 0]

13 Feb 2025

Please see our attached document 'Response to Reviewers' which includes a table of our responses to each editorial/reviewer comment.

---

## [Decision Letter · Decision Letter 1]

7 Mar 2025

Quality and costs of commissioned vs. researcher-led NIHR research: retrospective cohort study of randomised controlled trials

PONE-D-24-39387R1

Dear Dr. Hammond,

We’re pleased to inform you that your manuscript has been judged scientifically suitable for publication and will be formally accepted for publication once it meets all outstanding technical requirements. **Please consider the remaining suggestion of Reviewer #2 for the final version. **

Kind regards,

Sonia Vasconcelos, PhD

Academic Editor

PLOS ONE

Additional Editor Comments (optional):

Reviewers' comments:

Reviewer's Responses to Questions

**Comments to the Author**

1. If the authors have adequately addressed your comments raised in a previous round of review and you feel that this manuscript is now acceptable for publication, you may indicate that here to bypass the “Comments to the Author” section, enter your conflict of interest statement in the “Confidential to Editor” section, and submit your "Accept" recommendation.

Reviewer #1: All comments have been addressed

Reviewer #2: (No Response)

2. Is the manuscript technically sound, and do the data support the conclusions?

Reviewer #1: Yes

Reviewer #2: Yes

3. Has the statistical analysis been performed appropriately and rigorously? 

Reviewer #1: Yes

Reviewer #2: Yes

4. Have the authors made all data underlying the findings in their manuscript fully available?

Reviewer #1: Yes

Reviewer #2: Yes

5. Is the manuscript presented in an intelligible fashion and written in standard English?

Reviewer #1: Yes

Reviewer #2: Yes

6. Review Comments to the Author

Reviewer #1: The authors have adequately addressed the former comments at Discussion, including some additional topics for a more profound debate.

Reviewer #2: The authors have successfully addressed my comments and those of the other reviewer. My only suggestion now is to move the name of the statistical test from the title title to the column heading, in brackets after "p-value!.

7. PLOS authors have the option to publish the peer review history of their article (what does this mean? ). If published, this will include your full peer review and any attached files.

**Do you want your identity to be public for this peer review?** For information about this choice, including consent withdrawal, please see our Privacy Policy .

Reviewer #1: **Yes: ** Maria Gabriela Haye Biazevic

Reviewer #2: No

---

## [Editor Report · Acceptance letter]

PONE-D-24-39387R1

PLOS ONE

Dear Dr. Hammond,

I'm pleased to inform you that your manuscript has been deemed suitable for publication in PLOS ONE. Congratulations! Your manuscript is now being handed over to our production team.

Kind regards,

on behalf of

Dr. Sonia Vasconcelos

Academic Editor

PLOS ONE